# Monitoring Key Wheat Growth Variables by Integrating Phenology and UAV Multispectral Imagery Data into Random Forest Model

Shaoyu Han [1,2], Yu Zhao [2], Jinpeng Cheng [2], Fa Zhao [2], Hao Yang [2], Haikuan Feng [2], Zhenhai Li [2,3], Xinming Ma [1], Chunjiang Zhao [1,2] and Guijun Yang [2,*]

1   College of Agronomy, Henan Agricultural University, Zhengzhou 450046, China; 13598834331@stu.henau.edu.cn (S.H.); maxinming@henau.edu.cn (X.M.); zhaocj@nercita.org.cn (C.Z.)
2   Key Laboratory of Quantitative Remote Sensing in Agriculture of Ministry of Agriculture and Rural Affairs, Information Technology Research Center, Beijing Academy of Agriculture and Forestry Sciences, Beijing 100097, China; 2019201029@njau.edu.cn (Y.Z.); niceget@bjfu.edu.cn (J.C.); p18101005@stu.ahu.edu.cn (F.Z.); yangh@nercita.org.cn (H.Y.); fenghk@nercita.org.cn (H.F.); lizhenhai@sdust.edu.cn (Z.L.)
3   College of Geodesy and Geomatics, Shandong University of Science and Technology, Qingdao 266590, China
*   Correspondence: yanggj@nercita.org.cn

**Abstract:** Rapidly developing remote sensing techniques are shedding new light on large-scale crop growth status monitoring, especially in recent applications of unmanned aerial vehicles (UAVs). Many inversion models have been built to estimate crop growth variables. However, the present methods focused on building models for each single crop stage, and the features generally used in the models are vegetation indices (VI) or joint VI with data derived from UAV-based sensors (e.g., texture, RGB color information, or canopy height). It is obvious these models are either limited to a single stage or have an unstable performance across stages. To address these issues, this study selected four key wheat growth parameters for inversion: above-ground biomass (AGB), plant nitrogen accumulation (PNA) and concentration (PNC), and the nitrogen nutrition index (NNI). Crop data and multispectral data were acquired in five wheat growth stages. Then, the band reflectance and VI were obtained from multispectral data, along with the five stages that were recorded as phenology indicators (PIs) according to the stage of Zadok's scale. These three types of data formed six combinations (C1–C6): C1 used all of the band reflectances, C2 used all VIs, C3 used bands and VIs, C4 used bands and PIs, C5 used VIs and PIs, and C6 used bands, Vis, and PIs. Some of the combinations were integrated with PIs to verify if PIs can improve the model accuracy. Random forest (RF) was used to build models with combinations of different parameters and evaluate the feature importance. The results showed that all models of different combinations have good performance in the modeling of crop parameters, such as $R^2$ from 0.6 to 0.79 and NRMSE from 10.51 to 15.83%. Then, the model was optimized to understand the importance of PIs. The results showed that the combinations that integrated PIs showed better estimations and the potential of using PIs to minimize features while still achieving good predictions. Finally, the varied model results were evaluated to analyze their performances in different stages or fertilizer treatments. The results showed the models have good performances at different stages or treatments ($R^2 > 0.6$). This paper provides a reference for monitoring and estimating wheat growth parameters based on UAV multispectral imagery and phenology information.

**Keywords:** wheat growth variable; phenology; machine learning; random forest; UAV multispectral imagery

## 1. Introduction

Winter wheat is one of the most widely cultivated and fertilized food crops, and it is used in many products for human consumption [1]. Wheat growth monitoring is a

crucial part of achieving a reasonable yield, and field management is the following step after analyzing the crop growth situation. Among different kinds of field management, fertilizing has been recognized as the most important. Proper fertilizing is the key strategy to secure optimal crop yield. Nitrogen-based fertilizers provide vast N for crop growth. Nitrogen (N) takes part in multiple metabolisms and structural components, which makes N one of the most important elements in both crop and environmental sciences. It is an essential element in both wheat crop growth and yield formation [2]. While N deficiency makes it difficult to achieve the target yield, overfertilization is a common mistake in the unilateral pursuit of high yield. Excessive N applications lead to delayed maturity, which causes reduced yield, and adverse environmental impacts, such as soil contamination; furthermore, nitrogen is a major contributing source of greenhouse gas (GHG) [3,4]. Diagnosing crop growth status and variable rate fertilization can assist in avoiding the above problems. The principle of precision agriculture is the spatial and temporal variability of fertilizer. Therefore, the determination of crop status is the key procedure in practice [5]. Several parameters have been used for measuring the plant growth condition; for example, plant nitrogen concentration (PNC) and accumulation (PNA) are direct indicators of crop growth. Additionally, above-ground biomass (AGB) is another frequently used indicator because it is the proxy of the final yield. When the parameters are put together, the nitrogen nutrition index (NNI) is established by critical N dilution theory [6,7]. These parameters have been proved to be effective for use in variable rate fertilization; however, the issue is the speed of the process [8].

The timely and accurate monitoring of crop growth status is necessary for modern agricultural management. Traditionally, to acquire the growth variables, field samples are taken for lab analysis, which is time-consuming. Additionally, the results are spatially limiting. Thus, it is important to find a way to achieve more effective results [9]. Remote-sensing technology offers an alternative for assessing crop nutrient status, and crop parameters have been retrieved from remote-sensed data by different approaches [10]. With the emerging unmanned aerial vehicle (UAV) platform, which carries passive or active sensors, it is becoming easier to access rapid and non-destructive spatial results of crop growth parameters [11]. UAVs have advantages of flexibility and versatility; they are operated at relatively low cost while acquiring high spatial and temporal resolution data. In practice, the crop AGB can be estimated by RGB or multispectral images, and other N-related crop parameters can also be estimated by fusing image and spectral information. The reported modeling process includes seeking sensitive bands or VIs, and then using them to build models. Meanwhile, some studies investigated the optimal time window for growth monitoring [12–15]. These previous experiments have demonstrated the feasibility of UAV application.

Crop growth variables and retrieval methods can be categorized in three ways: empirical, physical, and hybrid methods. Previous research focused on the simple linear or non-linear relationships between vegetation indices (VI) and specific crop parameters [12,16], or used physical-based methods, known as radiative transfer models, to retrieve crop N status [17]. In order to make full use of the abundant UAV data, including band reflectance, VI or texture, and other features, machine learning regression algorithms of various kinds have been introduced for quantitative vegetation remote sensing [18–20]. Machine learning methods are becoming powerful modelling tools to interpret information from large amounts of remotely sensed data. Among different regression algorithms, random forest (RF) is a classic and powerful method [21]. It is an ensemble learning model that combines a large number of decision trees, which makes it robust when the model consists of many input variables. RF models have been widely used in crop classification, growth monitoring, and yield forecast [22,23]. Additionally, RF prevails in the previous comparison studies of different algorithms for monitoring different crops [24,25]. These studies have yielded satisfactory results by using RF models, both in classification and prediction, which strongly illustrated the feasibility of the RF model.

Although crop parameters are estimated by different techniques using remotely sensed data, a common problem is that these kinds of models neglect the fact that crops are significantly different in their different growth stages. Crop growth is an allometry process. The morphology traits of a crop can change substantially from the vegetation to reproductive stages, and the leaves, stems, and spikes play different roles in different stages. This would cause significant impacts on remote-sensing observations. In the practice of crop vegetation remote sensing using optical sensors, the leaves, stems, and panicles are the spectrally responsive organs. Leaves are always the major sources of reflection in the different stages [26]; however, as the growth stages progress, leaves will transform from a sink to a source of assimilates. When leaves are sinks, their major function is as the storage warehouse of photosynthesis production, while they turn into a photosynthesis producer when the leaves are mature. At the same time, the stems start to become the major storage of the assimilates, especially after the shift from the vegetation to the reproductive stage, and the panicles become the new sink for the yield formation. During this period, the characters of the leaves, stems, and panicles keep changing, affecting sink and flow interrelation and transformation; consequently, biomass and N deposits will translocate in different sinks accordingly [27], id est, the leaves will provide substantial reflectance while not always being responsible for the majority of crop biomass and nitrogen storage. Since this asymmetric information is hard to be exhibited in spectral information [28], it might explain the deficits of the simple linear model; furthermore, this phenomenon suggests that phenology information should be considered in the crop monitoring models. Several researchers have pointed out that crop phenology is important in predicting crop growth conditions or forecasting yield [29,30].

Current machine learning models for monitoring crop growth status have been established by solely using multiple vegetation indices [12,31]. In addition to this, efforts include fusing color features or combining them with cultivar information [32,33]. Moreover, canopy fluorescence is an increasingly popular technique that can be used for growth monitoring [34]. Furthermore, using spectral-based deep learning is also a rational approach [35]. These studies utilized spectrum information in addition to other vegetation characteristics. Only a few studies considered the variation of phenology and integrated it into the model. Since the crop growth status significantly varied from stage to stage, optical sensors had a limited ability to track this inherent variation. The addition of different types of data could be descriptive for different stages, which clearly showed that the direct application of phenology is appealing in the context of adjusting the model instability in multi-stage scenarios. Therefore, the main objectives of this study are:

(1) To verify the phenology effect on retrieving wheat crop parameters from UAV multi-spectral data;
(2) To use RF models to evaluate the accuracy of different combinations of band reflections, Vis, and PIs in wheat parameters prediction;
(3) To Specify the accuracy of different growth stages and N treatments in established models to understand the model applicability.

## 2. Materials and Methods

### 2.1. Experimental Site and Design

We conducted the experiment during the 2020–2021 winter wheat growing season at the Xiaotangshan National Experiment Station for Precision Agriculture (116°26′36″ E, 40°10′44″ N) in Beijing, China (Figure 1). The field site is in the northernmost section of the North China Plain, which has a temperate monsoon semi-humid climate of medium latitudes, with an average altitude of 36 m. It has an average annual precipitation of 500–600 mm and an average annual temperature of 12 °C. The annual amount of solar radiation is 4800 MJ m$^{-2}$ (China Meteorological Data Service, http://data.cma.cn/ accessed on 6 January 2022). We provide the weather information of the experiment duration in Figure 2.

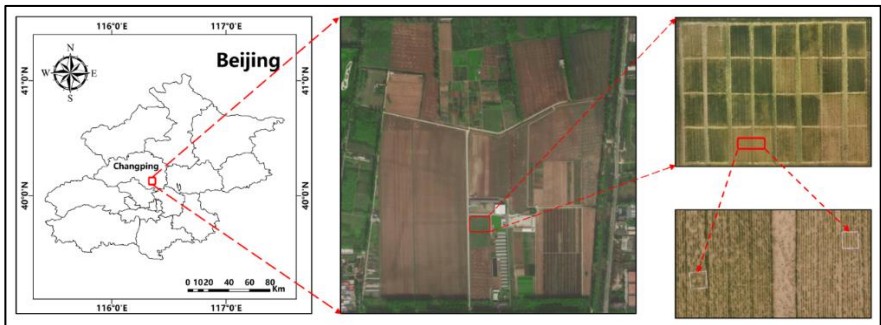

**Figure 1.** Location of Xiaotangshan National Experiment Station for Precision Agriculture and layout of experiment plots.

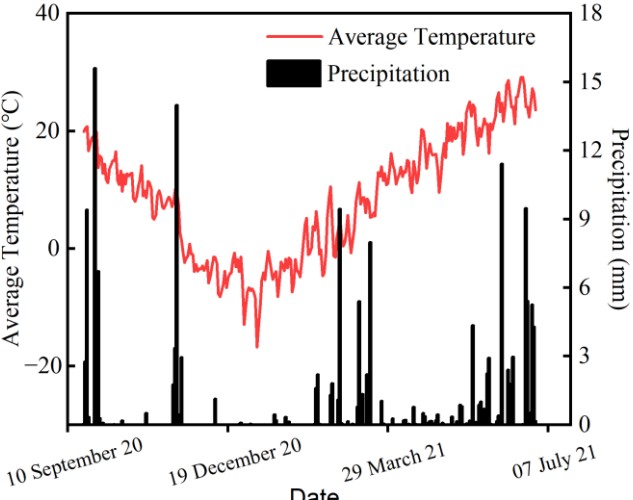

**Figure 2.** Average temperature and precipitation during growing season.

This study was part of an ongoing long-term fertilizer experiment. We selected two local wheats (*Triticum durum* L. cultivar. JH11 and cultivar. ZM1062) and four nitrogen fertilizer rates (0, 90, 180, 270 N/ha) in the field experiment. The plot size was $9 \times 15$ m. We set row spacing to 15 cm, and we uniformly set the plant density to $360 \times 10^4$ plants ha$^{-1}$. We settled treatments in all field experiments using a complete randomized block design with four replicates. We divided the nitrogen fertilizer application into two splits and applied 1:1 for the base and top-dress before sowing and at the jointing stage.

The primary soil type is a clay loam soil by Food and Agriculture Organization (FAO) soil classification, with a pH of 7.7, 19 g·kg$^{-1}$ organic matter, 1.01 g·kg$^{-1}$ total N, 14.5 mg·kg$^{-1}$ Olsen-P, and 127.9 mg·kg$^{-1}$-available K in the 0–20 cm-surface soil layer. We performed other field management procedures, including weed control, pest management, and our application of phosphate and potassium fertilizer followed local standard practices for winter wheat production.

### 2.2. Crop Data Acquisition and Calculation of Nitrogen Nutrient Index

We conducted five experiments in five key wheat growth stages (jointing, booting, anthesis, early filling, late filling). There were 24 samples in the jointing stage and 32 in the other stages. In total, the sampling number was 152. During field sampling at each stage, we randomly collected 20 tillers around the white frame of each plot. We immediately took fresh samples to the laboratory and separated them into leaves and stems, and also spikes after they emerged. We put the samples into paper bags and placed them in the oven at 105 °C for 20 min to stop metabolism, and then dried at 80 °C until the samples became constant weight. We recorded the dry weight of each sample by a balance with an accuracy of 0.001 g. After we acquired the AGB, we analyzed all samples for N concentration using

the micro-Kjeldahl method. We calculated the total plant N concentration as the ratio of the total N accumulation to AGB.

$$AGB = \frac{(LW + SW + PW)\cdot T}{20\cdot L} \tag{1}$$

$$PNA = (LW\cdot LN + SW\cdot SN + PW\cdot PN) \tag{2}$$

$$PNC = \frac{PNA}{AGB} \tag{3}$$

where *LW*, *SW*, and *PW* are the dry weights of leaf, stem, and panicle samples, respectively. *LN*, *SN*, and *PN* are the N concentrations of leaf, stem, and panicle samples, respectively. *T* is the number of winter wheat stems per unit area and L is the row spacing (15 cm). Subsequently, PNA is plant nitrogen accumulation and PNC is plant nitrogen content.

As described by Lemaire [36], we calculated the nitrogen nutrition index (NNI) of each treatment within various growth stages by the following equation:

$$NNI = N_a / N_c \tag{4}$$

where the $N_a$ represents the actual N concentration, $N_c$ represents the critical N concentration. The $N_c$ curve used in this study was adopted from previous research [37]:

$$N_c = 5.35\cdot AGB^{-0.53} \tag{5}$$

We established the equation from a previous nitrogen fertilizer experiment in the same field. Crop nitrogen status is normal when NNI is between 0.95 and 1.05, it overdoses when NNI exceeds 1.05, and there is a deficit when it is less than 0.95.

### 2.3. Acquisition and Preprocessing of UAV Images

We used a DJI Phantom 4 Multispectral 4-rotor-wing unmanned aerial vehicle (UAV) (DJI-P4M, SZ DJI Technology Co., Ltd., Shenzhen, China) to capture multispectral images. The UAV had 2 million pixel multispectral sensors consisting of 6 cameras, including Blue (450 nm $\pm$ 16 nm), Green (560 nm $\pm$ 16 nm), Red (650 nm $\pm$ 16 nm), RE (730 nm $\pm$ 16 nm), NIR (840 nm $\pm$ 26 nm), and visible light (RGB). The details of the UAV and sensor information are in Figure 3.

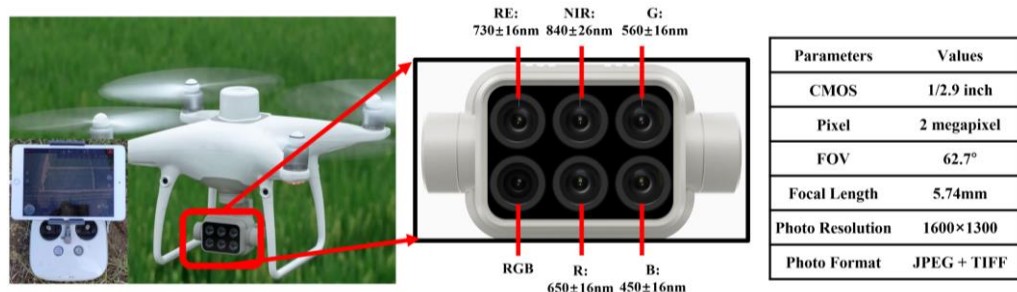

**Figure 3.** DJI P4M and sensor properties used in study.

We conducted five flights within each key wheat growth stage. We set the flight mission height to 30 m, with a speed of 4 m/s. We set the image overlap and sidelap to 80%. The ground spatial resolution is 1.6 cm under such parameters. We performed all flight missions between 10:00 and 12:00 on clear and cloudless days. Prior to each flight, we collected calibration images with a standard reflectance panel. The panel is a fine cloth installed inside a plastic box, and it has a basic reflectance of 0.797, 0.872, 0.877, 0.875, and 0.867 for Blue, Green, Red, RE, and NIR, respectively. We calibrated each image to follow the specific band. After the flight, we calibrated our collected multispectral images and processed into ortho-mosaic maps using DJI Terra software (Terra, SZ DJI Technology Co., Ltd., Shenzhen, China) for further analysis.

### 2.4. Feature Extraction and Determination

We used the ortho-mosaic maps for band reflectance and VIs extraction. We extracted all data within the white frames using shapefiles for each stage. Additionally, we calculated plot average of all pixel values as the extraction results.

We used three types of features in this study, including the original multispectral band reflectance as the first type, vegetation indices for the second type, and phenology indicators (PIs) for the third type. For the first two types of data, because we extracted them from UAV images, in order to keep consistency, we acquired all of them from within the white frame (Figure 1). We collected the original reflectance of each band as the first data type, and then we selected several vegetation indices as second data type, of which we used part of them to model AGB. Additionally, we used part of them to retrieve key growth parameters in the previous studies, which indicated all the selected VIs have reasonable potential to be used for crop nitrogen status monitoring. As for the third type, we recorded the five stages that represent the typical wheat growth process as Zadok growth stages—phenology indicators. In this case, ZS33, ZS47, ZS65, ZS75, and ZS80 represent the wheat jointing, flag leaf, anthesis, early filling, and late filling stages, respectively. In order to maintain data uniformity, we simplified the phenology into a number of stages. The three types of data are listed in Table 1.

**Table 1.** List of used features in this study.

| Data Type | Features | Acronym | Equation | Reference |
|---|---|---|---|---|
| Original Reflectance | Blue Band Reflectance | B | Reflectance of B band | / |
| | Green Band Reflectance | G | Reflectance of G band | / |
| | Red Band Reflectance | R | Reflectance of R band | / |
| | RedEdge Band Reflectance | RE | Reflectance of RE band | / |
| | Near-Infrared Band Reflectance | NIR | Reflectance of NIR band | / |
| Vegetation Indices | Difference Vegetation Index | DVI | $NIR - R$ | [38] |
| | Enhanced Vegetation Index | EVI | $2.5 \times \dfrac{NIR - RED}{NIR + 6 \times R - 7.5 \times B + 1}$ | [39] |
| | Enhanced Vegetation Index 2 | EVI2 | $2.4 \times \dfrac{NIR - R}{NIR + R + 1}$ | [40] |
| | Leaf Chlorophyll Index | LCI | $\dfrac{NIR - RE}{NIR + R}$ | [41] |
| | Modified Chlorophyll Absorbtion Ratio Index | MCARI | $\dfrac{((RE - R) - (0.2 \times (RE - G))) \times RE}{R}$ | [38] |
| | Modified Non-Linear Index | MNLI | $\dfrac{1.5 \times NIR^2 - 1.5 \times G}{NIR^2 + R + 0.5}$ | [42] |
| | Modified Soil-Adjusted Vegetation Index | MSAVI | $\dfrac{2 \times NIR + 1 - \sqrt{(2 \times NIR)^2 - 8 \times (NIR - RED)}}{2}$ | [43] |
| | Modified Simple Ratio Index | MSR | $\dfrac{\frac{NIR}{R} - 1}{\sqrt{\frac{NIR}{R} + 1}}$ | [44] |
| | Normalized Difference Red-Edge | NDRE | $\dfrac{NIR - RE}{NIR + RE}$ | [45] |
| | Normalized Difference Vegetation Index | NDVI | $\dfrac{NIR - R}{NIR + R}$ | [46] |
| | Ratio Vegetation Index | RVI | $\dfrac{NIR}{R}$ | [47] |
| | Soil-Adjusted Vegetation Index | SAVI | $\dfrac{NIR - R}{NIR + R + 0.5} \times (1 + 0.5)$ | [48] |
| Phenology Indicators | Phenology Indicators | PI | 33 (jointing stage), 47 (flag leaf stage), 65 (anthesis), 75 (early filling), 80 (late filling) | [49] |

### 2.5. Data Analysis and Model Establishment

We analyzed the 4 crop parameters data by Tukey's HSD test to distinguish the differences across 5 stages. We performed a three-way analysis of variance (ANOVA) to explore how much the phenology contributes to the biometrics. We analyzed the effects of cultivars, N treatment, and phenology on AGB, PNA, PNC, and NNI. We used Duncan's test to analyze differences in parameter averages between treatments. The threshold for statistical significance was $p < 0.05$.

In this study, we employed RF to build models for AGB, PNA, PNC, and NNI. As mentioned above, we grouped all the features into three types of data, and we selected these features as six combinations (Table 2).

**Table 2.** Combinations of features in RF models.

| | **Band** | **VI** | **PI** |
|---|:---:|:---:|:---:|
| C1 | ✓ | | |
| C2 | | ✓ | |
| C3 | ✓ | ✓ | |
| C4 | ✓ | | ✓ |
| C5 | | ✓ | ✓ |
| C6 | ✓ | ✓ | ✓ |

RF is an ensemble technique that combines multiple decision trees, and each tree in the forest predicts independently. We put all predictions into a vote to make the final prediction. It is a practicable model when dealing with small sample sizes. RF models involve a hyperparameter-adjusting process to maximize the accuracy of the models. Thus, we optimized the models' hyperparameters through 10-fold cross-validation. The only parameter in random forests that typically need optimization are the number of trees in the ensemble. We settled on a total of 200 decision trees for the model based on the stable results from primary validation. We also used the number of decision trees in related remote-sensing studies [50].

To analyze the special effects of our most interesting PIs we performed multiple iterations to deconstruct the model feature by feature. Firstly, we ranked each feature based on the Bayesian framework; therefore, the ranking result can be used as a feature reduction tool. Then, we removed the lowest relevant feature in each iteration. Eventually, we found the most sensitive features. Through iterations, we determined the minimum number of features to build a model while keeping the accuracy acceptable.

### 2.6. Model Evaluation

We built and evaluated all models by 10-fold cross-validation, and we used the mean results of cross-validation in the model comparisons. We used three commonly used indices ($R^2$, RMSE, and NRMSE) to compare the performance of generated models The calculation equations of $R^2$, RMSE, and NRMSE are as follows:

$$R^2 = 1 - \frac{\sum_{i=1}^{n} (y_i - y_i')^2}{\sum_{i=1}^{n} (y_i - \overline{y})^2} \tag{6}$$

$$RMSE = \sqrt{\frac{\sum_{i=1}^{n} (y_i - y_i')^2}{n}} \tag{7}$$

$$NRMSE = \frac{RMSE}{N} \tag{8}$$

where $y_i$ and $y_i'$ are the measured and predicted values for sample $i$, respectively. $\overline{y}$ indicates the mean values and $n$ is the number of samples used for calibration or validation set. $N$ is the average value of the samples.

## 3. Results

### 3.1. Statistics of Crop Data and Their Relationships with Selected Bands and VIs

Field experiment results of the four parameters are summarized in Figure 4. Across the different stages, the AGB and PNA increased with a steady overall trend; however, noticeably, their growth rates peaked at different stages. AGB had rapid growth rates at different stages, e.g., ZS47~ZS65, and they were significantly different because of the start of the filling stages. PNA grew fast in the early stages of ZS33~ZS47 and they showed significant differences, indicating that plants absorb plenty of nitrogen at the jointing stage. As for PNC, it reached stabilization after decreasing at the early stages. PNC showed a significant downward trend in the early stages, especially ZS33 and ZS47. The NNI trend through the stages had little fluctuations for the various nitrogen treatments in this study, which shows it is a promising indicator for evaluating wheat N status.

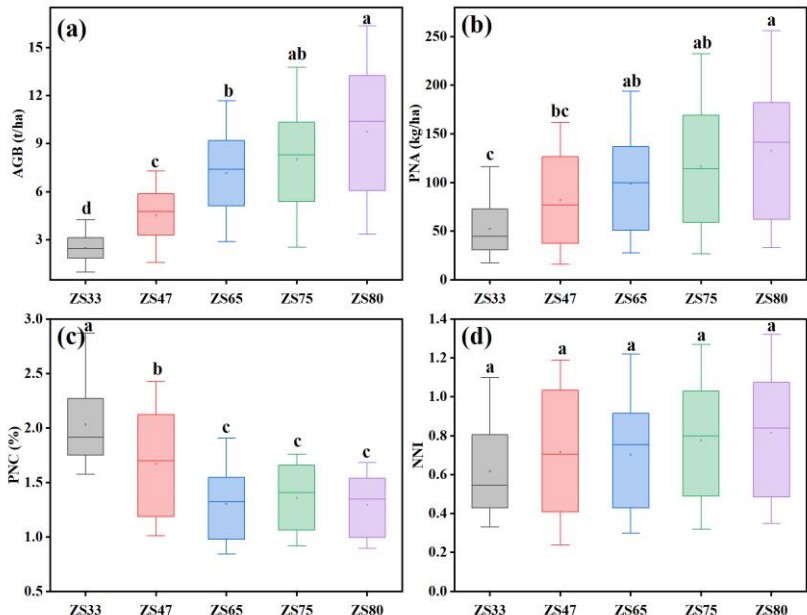

**Figure 4.** Wheat growth data collected at five stages: (**a**) AGB, (**b**) PNA, (**c**) PNC, (**d**) NNI. The letters on top of box are significance marks.

Linear regressions were performed between biometrics and the reflectance of different bands, and the Red reflectance had the best relationship with all biometrics. Then, Green and Blue had relatively strong relationships with AGB, PNA, and NNI. For NIR, it had good relationships with PNA, PNC, and NNI. RE showed low relativity with the biometrics. From another view, PNC is hard to model from band reflectance, but the other three are comparatively easier. The highest $R^2$ was 0.42 between the Red band reflectance and NNI.

Regression between 12 selected VIs and four biometrics is recorded in Table 3. Almost all the VIs were closely related to the biometrics, and the best determination coefficient was made by using NDRE for AGB, with an $R^2$ of 0.64; overall, the VIs had the best relationships (0.34~0.63) with NNI, and were greatly related to AGB (0.1~0.64) and PNA (0.29~0.58), while they had a relatively lower $R^2$ (0.01~0.39) between VIs and PNC.

**Table 3.** Coefficient of determination ($R^2$) between biometrics and reflectance or vegetation indices.

|  | Feature | AGB | PNA | PNC | NNI |
|---|---|---|---|---|---|
| Original Reflectance | Blue | 0.21 ** | 0.17 ** | 0.00 ns | 0.13 ** |
|  | Green | 0.22 ** | 0.25 ** | 0.08 ** | 0.26 ** |
|  | Red | 0.25 ** | 0.35 ** | 0.19 ** | 0.42 ** |
|  | RE | 0.11 ** | 0.07 ** | 0.00 ns | 0.03 * |
|  | NIR | 0.03 * | 0.14 ** | 0.21 ** | 0.24 ** |
| Vegetation Indices | DVI | 0.16 ** | 0.35 ** | 0.35 ** | 0.52 ** |
|  | EVI | 0.1 ** | 0.29 ** | 0.39 ** | 0.48 ** |
|  | EVI2 | 0.38 ** | 0.5 ** | 0.06 ** | 0.52 ** |
|  | LCI | 0.28 ** | 0.33 ** | 0.08 ** | 0.34 ** |
|  | MCARI | 0.21 ** | 0.37 ** | 0.29 ** | 0.49 ** |
|  | MNLI | 0.22 ** | 0.43 ** | 0.33 ** | 0.59 ** |
|  | MSAVI | 0.23 ** | 0.45 ** | 0.36 ** | 0.61 ** |
|  | MSR | 0.31 ** | 0.52 ** | 0.29 ** | 0.63 ** |
|  | NDRE | 0.64 ** | 0.58 ** | 0.01 ns | 0.43 ** |
|  | NDVI | 0.3 ** | 0.49 ** | 0.32 ** | 0.61 ** |
|  | RVI | 0.3 ** | 0.5 ** | 0.26 ** | 0.6 ** |
|  | SAVI | 0.23 ** | 0.45 ** | 0.36 ** | 0.61 ** |

Significance level: ns = not significant, * $p < 0.05$, ** $p < 0.01$.

### 3.2. Phenology Contribution in Estimating Crop Data

The results of Section 3.1. showed the best performing model was built by R reflectance. Therefore, it was selected along with the commonly used NDVI to plot Figures 5 and 6. They showed the linear relationships of single factors with biometrics across different stages, and we noticed that there were clear phenology differentials. In Figure 5a, at different stages, the slope and intercept vary from −72.42~ to −18.56 and 4.33 to ~16.57, respectively. We show similar gaps in Figure 5b,c; however, in Figure 5d, the slope and intercept change slightly at different stages. Figure 5 shows it is inferable that the VIs have similar results, such as in NDVI and the drastic slope and intercept variations of AGB, PNA, and PNC; however, NNI shows little change. There is an apparent negative correlation between R band reflectance and crop parameter and a positive correlation between NDVI and crop parameters. This situation is because as crop growth increases, red light is increasingly absorbed by crop vegetation, with the NIR band showing a higher reflection rate. At late stages, mature crops tend to show low reflection rates in both bands.

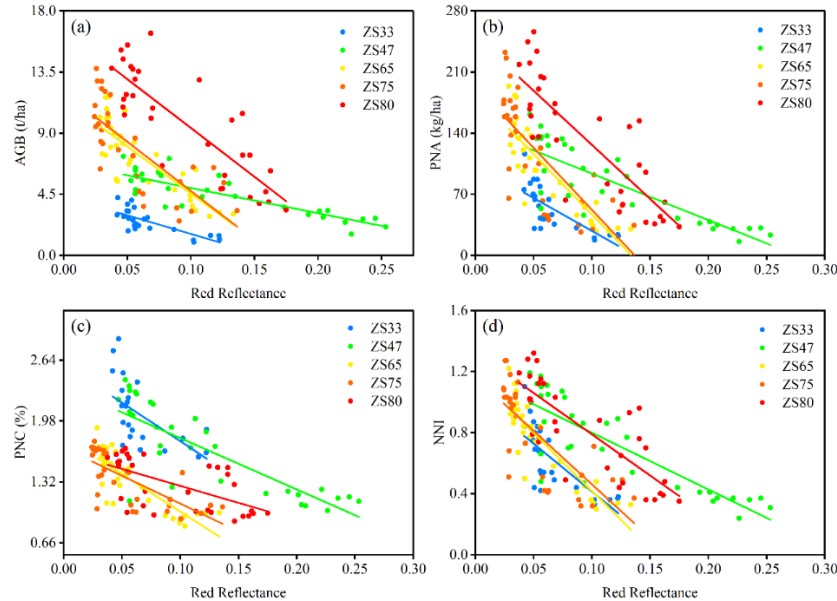

**Figure 5.** Linear relationships between R-band reflectance and crop parameters at all phenologies: (**a**) AGB, (**b**) PNA, (**c**) PNC, (**d**) NNI.

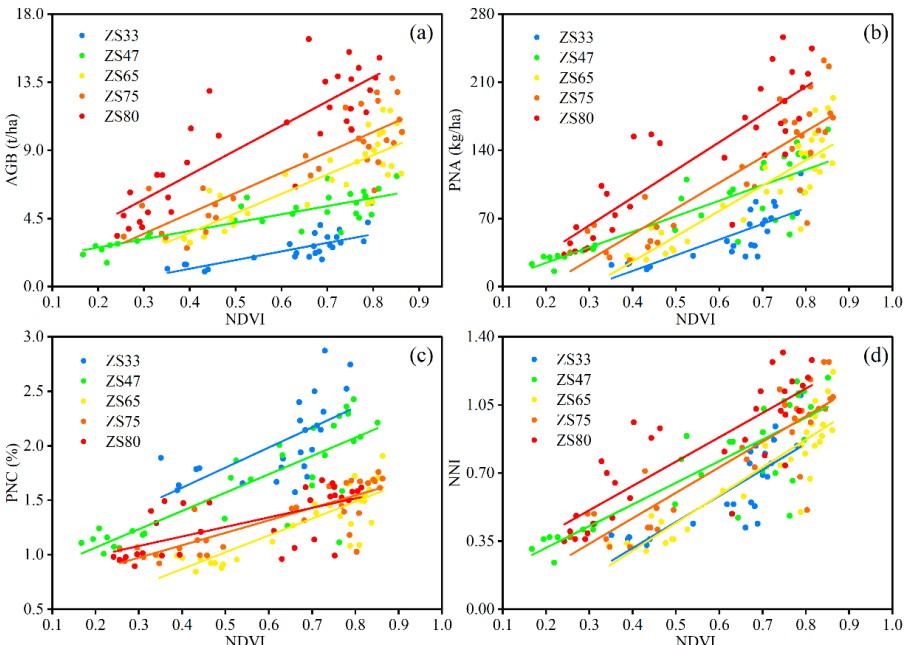

**Figure 6.** Linear relationships between NDVI and crop parameters at all phenologies: (**a**) AGB, (**b**) PNA, (**c**) PNC, (**d**) NNI.

Nitrogen treatment and phenology are the main factors affecting the four parameters, as shown in the three-way ANOVA (Table 4). Among the results, nitrogen treatment contributed mostly in NNI (72.64%), and the least in AGB (36.81%). Phenology contributed majorly in AGB and PNC but minorly in PNA (35.02%), and it contributed even less in NNI (16.04%). Besides the main factors, the interactions between nitrogen fertilizer and phenology also showed significant effects (5.88~12.22%) on the biometrics. From a statistical view, the ANOVA results strongly suggested that phenology plays an important role in modeling crop parameters.

**Table 4.** A three-way ANOVA for effects of wheat variety (V), nitrogen fertilizer (N), and phenology (P) on AGB, PNA, PNC, and NNI.

| Factor | AGB | | PNA | | PNC | | NNI | |
|---|---|---|---|---|---|---|---|---|
| | *p* Value | Contribution | *p* Value | Contribution | *p* Value | Contribution | *p* Value | Contribution |
| V | 0.941 | 0.00% | 0.654 | 0.15% | 0.114 | 1.22% | 0.428 | 0.59% |
| N | 0.000 | **36.81%** | 0.000 | **51.53%** | 0.000 | **41.54%** | 0.000 | **72.64%** |
| P | 0.000 | **49.87%** | 0.000 | **35.02%** | 0.000 | **40.22%** | 0.001 | **16.04%** |
| V*N | 0.819 | 0.64% | 0.616 | 1.35% | 0.237 | 2.04% | 0.593 | 1.75% |
| V*P | 0.990 | 0.21% | 0.975 | 0.36% | 0.963 | 0.29% | 0.946 | 0.69% |
| N*P | 0.123 | 10.96% | 0.287 | 9.78% | 0.004 | 12.22% | 0.873 | 5.88% |
| V*N*P | 0.999 | 1.52% | 0.998 | 1.80% | 0.947 | 2.45% | 0.997 | 2.42% |

*3.3. RF Model Results of Different Combinations*

3.3.1. Model Results Using All Features

The models were built for crop parameters according to the combinations mentioned in Table 2. In each combination, all features were used to build models. The results are in the heatmap of Figure 7, which is a comparison between different combinations. Generally, combinations with more features performed better. Figure 7a shows the data of $R^2$ increase from C1 to C6. Figure 7b,c show it decreases from C1 to C6. The model with the poorest predictions was C1, which only uses band reflectance. The best results were C5 for PNC, with $R^2 = 0.77$, RMSE = 0.2, and NMRSE = 10.32, and C5 for NNI, with $R^2 = 0.73$, RMSE = 0.15, and NMRSE = 13.92, while C6 for AGB and PNA had the highest $R^2$ and lowest RMSE or NRMSE.

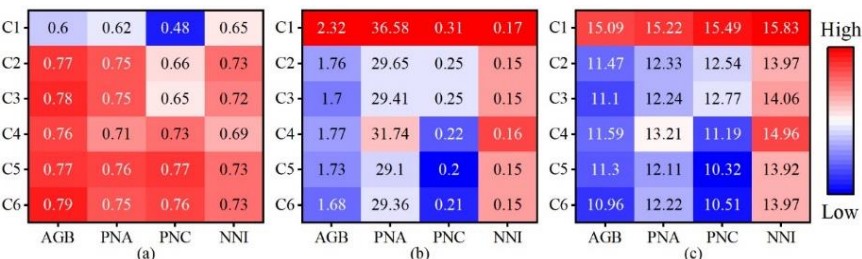

**Figure 7.** RF modeling results for different combinations: (**a**) $R^2$, (**b**) RMSE, (**c**) NRMSE.

Compared with C1, C2, and C3, the C4, C5, and C6 include PIs as part of their features, and the latter showed better estimations. In a comparison between C1 and C4, the PIs significantly improved the model that only used band reflectance for four crop parameters, such as $R^2$ from 0.48~0.65 to 0.69~0.76 or NMRSE from 15.09~15.83 to 11.19~14.96. This showed that PIs had the potential to enhance the models of AGB, PNA, and PNC. Furthermore, in the comparison between C2 and C5, PIs only improved the VI model in predicting PNC ($R^2$ from 0.66 to 0.77), and C6 performed better than C3 or C5, showing the best results using all features.

Overall, all models showed good estimations of the parameters, particularly C5 and C6, which showed great results for all crop parameters. C5 was the combination of VI and PI, and C6 was the combination of all three types of data. The best model for AGB was C6, and the best model for PNA, PNC, and NNI was C5. The results showed that AGB, PNA, and PNC were better estimated when integrated with PIs; however, NNI showed insensitivity to PIs.

### 3.3.2. Model Iteration and Feature Selection Results

In Figure 7, we show the integration of the C4, C5, and C6 models with PIs, and they showed a generally high performance, especially C6, which had the highest $R^2$ (0.79). The comparisons between the results of C1 and C4 and C3 and C6 showed that phenology might play an important role in predicting key wheat growth parameters. Therefore, we disassembled the model to comprehend whether the models were enhanced by coupling phenology indicators. It was identified that C5 and C6 were the best models for four crop parameters because they had similar prediction abilities and C6 took all features in the model. Consequently, a feature-by-feature iteration was performed for C6, and the results are presented in Figure 8.

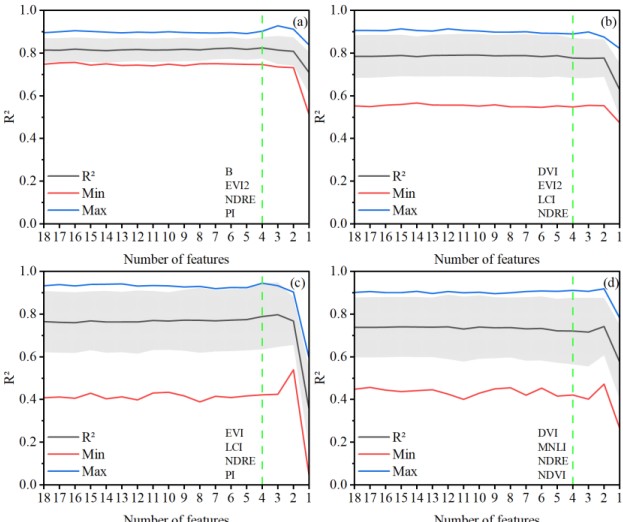

**Figure 8.** $R^2$ of model iteration in the best model: (**a**) AGB, (**b**) PNA, (**c**) PNC, (**d**) NNI; gray area, red line, and blue line represent the SD, min, and max of 10-fold cross-validation, respectively. The green dotted lines were models with 4 features, and texts along them are features of this particular model.

Of all crop parameter models presented in Figure 8, AGB and PNA predicted with fewer error ranges, while PNC and NNI had apparent error ranges. All models tended to show that stable estimations were acquired at more than three features. Therefore, we selected the model with four features for further analysis, and the details of four-feature models are presented in the text of the figures. The AGB and PNC models were integrated with PIs, while the PNA and NNI models were built by VIs. Overall, EVI2, NDRE, and PIs had the most appearances. However, the highest mean $R^2$ varied when using a different number of features. For AGB models, $R^2$ was the highest at four features, and they were B, EVI2, NDRE, and PIs. Additionally, in Figure 8c for PNC, three features of EVI, LCI, and PIs had the highest $R^2$. For PNA models, although the model seemed stable at minimum features, the best $R^2$ was at 11 features. The best models for NNI were built with two VIs of MNLI and NDRE. It is worth mentioning that PIs took great priority in the AGB and PNC models because the best model emerged with the appearance of PIs; however, PIs showed little significance in the PNA and NNI models.

We present the relative importance of different variables in Figure 9. The number of features in different parameters was set to four. In the models of AGB, PNA, PNC and NNI, and NDRE, the relative importance was 46.8%, 39.9%, 11.4%, and 42.7%, respectively. Our interested PIs had a relative importance of 10.1% and 47.9% in AGB and PNC, respectively. These results showed that PIs provide a good contribution in the model performances of AGB and PNC. Especially in PNC, PIs showed the highest importance among all features. Taken together in Figures 8 and 9, it was obvious that PIs can significantly improve the model accuracy in few-parameter circumstances.

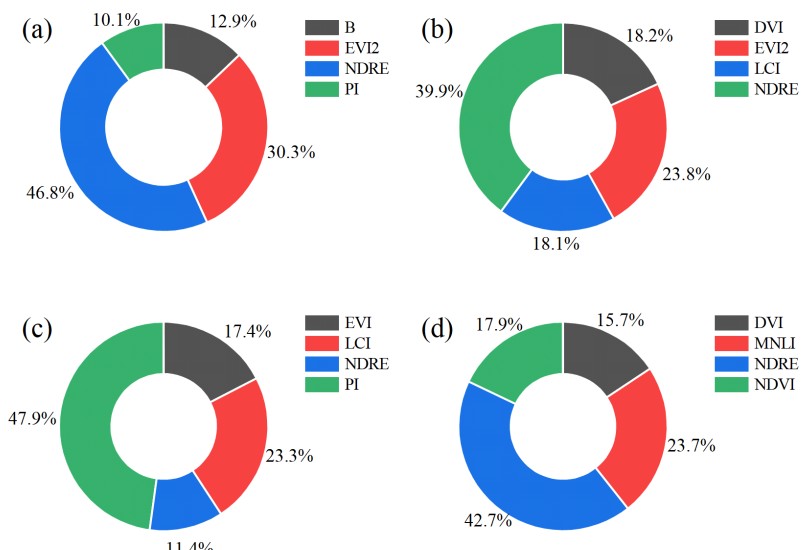

**Figure 9.** Relative importance of different variables in best models (%): (**a**) AGB, (**b**) PNA, (**c**) PNC, (**d**) NNI.

*3.4. Model Validation and Spatial Results of UAV Data*

After the iterations and best features were found, the model for the four parameters was evaluated using all data. The results we present in Figure 10 show that all parameters' models showed great estimation accuracy. Among the four parameters, AGB and PNC were better estimated ($R^2 > 0.80$), and NNI had the lowest $R^2$ of 0.74. These results showed the model was effective and demonstrated the feasibility of integrating PIs into the machine learning model.

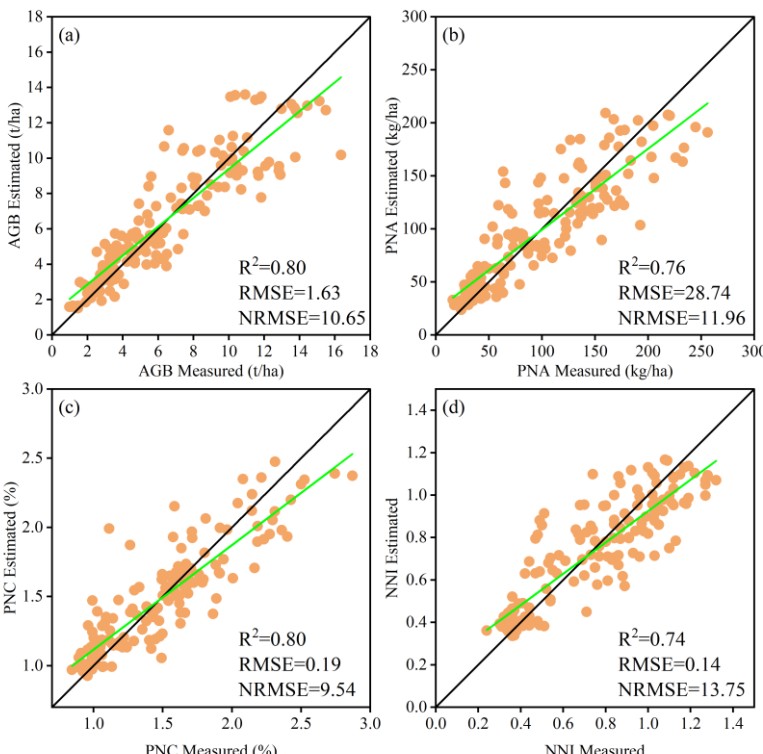

**Figure 10.** Model evaluation using all data. (**a**) AGB, (**b**) PNA, (**c**) PNC, (**d**) NNI. The 1:1 lines and fitting curves were plotted as black and green lines, respectively.

Among all models built in this study, the best models were used to build pixel-level spatial results of the four wheat parameters. The results of one key-stage anthesis (ZS65) are in Figure 11. The spatial results were highly consistent with the field experiment. The inside-plot variance of each treatment is low, and the difference between different-plot treatments is obvious. In this manner, the results were acquired ready for field-level precision fertilizing.

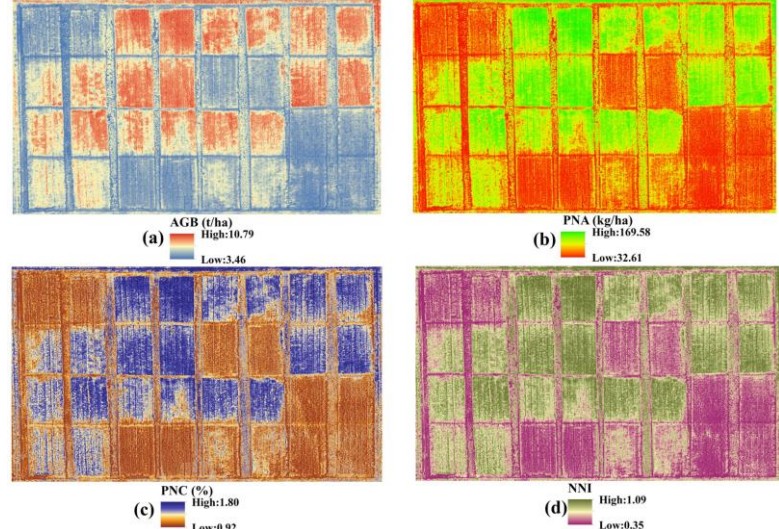

**Figure 11.** Spatial results of four crop parameters using best models at ZS65: (**a**) AGB, (**b**) PNA, (**c**) PNC, (**d**) NNI.

### 3.5. Model Accuracy in Different STAGES and N treatments

A further analysis was performed to specify how the model accuracy varied in different stages or N treatments, and all predication results from 3.4 were extracted and grouped by stages or N treatments. The stage results are presented in Figure 12. PNA, PNC, and NNI models showed higher $R^2$ and lower RMSE at different stages, with an average $R^2$ of 0.72, 0.70, and 0.75, respectively, and AGB showed a relatively lower average $R^2$ of 0.61 and a higher RMSE. As the growth period advanced, the trends of the four parameters were different. The PNA and NNI models showed steady results, the AGB model had better estimations in the late stages, and conversely, the PNC model showed lower results in the late stages. Among all stages, ZS65 generally had high model accuracy.

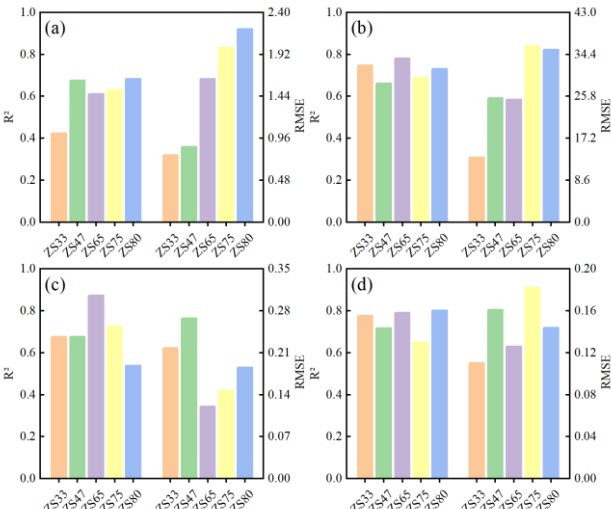

**Figure 12.** Model accuracy comparison at different stages: (**a**) AGB, (**b**) PNA, (**c**) PNC, (**d**) NNI.

Similarly, Figure 13 shows another analysis that was performed in a different N treatment. The models of AGB and PNC showed higher $R^2$, while the PNA and NNI models showed lower performances at different N treatments. For all N treatments, N0, N2, and N3 showed an average $R^2$ of 0.63, 0.68, and 0.68, respectively. The AGB model of N1 treatment showed great results; however, the other three models did not perform well in N1. Generally, the AGB model showed good results in all N treatments and the other models showed good performances in high N treatments. These results clarified that the models we built in this study have different accuracies among different stages and N treatments. This means the PIs can have different impacts on the model accuracy in terms of the experiment setup.

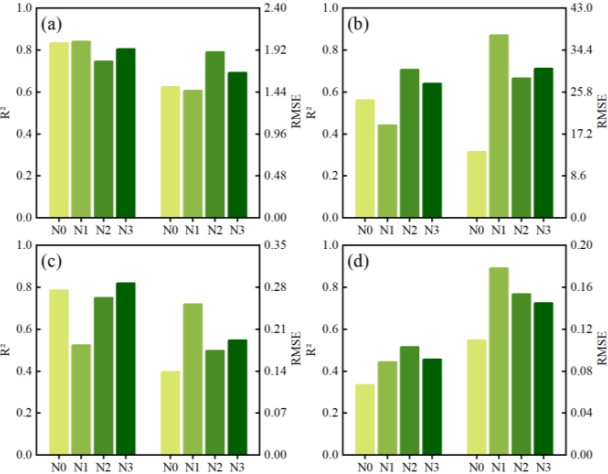

**Figure 13.** Model accuracy comparison in different N treatments: (**a**) AGB, (**b**) PNA, (**c**) PNC, (**d**) NNI.

## 4. Discussion

### 4.1. Comparison between Models Using Band or VI

In this paper, we firstly evaluated the linear model built from bands and VIs. Generally, band reflectance models performed less effectively compared with the VI models, which is easy to understand because VI is the synthetic of multiple bands' information; therefore, it can better perform than a single band. When putting more bands into consideration, the model performance is improved. Similar results were found in using bands' information to model the rice leaf area index (LAI) [51]. As shown in Figures 5 and 6a, a common situation is that the band or VI is easily saturated in the late growth stages due to high vegetation cover or biomass. The possible explanation is that the R band cannot penetrate deeper while the canopy is densely closed, and so are bands within the visible light. On the contrary, RE is less absorbed by the upper canopy, which means it can penetrate further into the canopy and carry more canopy information [13,52]. This physical limitation could be the major cause of the lower accuracy of models using bands.

In this paper, RF is also used to establish band and VI models to predict wheat growth status parameters. The model accuracy is significantly improved by only using band information. As the results in Figure 7 showed, C1 had the lowest accuracy. C2 and C3 showed great prediction abilities. The results of the saturation problem can be alleviated in RF models; meanwhile, in our models, NDRE and LCI were selected as features to predict AGB, PNC, and NNI, which could be a reasonable outcome because they were the indices using the RE band [12,53].

### 4.2. Integrating PIs into Crop Growth Monitoring Is Promising

At present, research has been performed to use texture information [8,54], RGB color features [32,55], crop height [56], meteorological factors, or soil data [57] as variables to build models for crop monitoring, along with other deep learning approaches, such as convolutional neural networks (CNNs) [58,59]. These methods successfully built the model for crop status monitoring; however, the models can be complicated and redundant. As a matter of fact, optical sensors tended to show stable variations of the reflectance gradient in one particular experiment; however, in several experiments across different growth stages, the stable outcome is affected by many factors, such as major plant growth status, soil conditions, and the interactions of these factors [30]. Hence, it is necessary to add phenology information in order to address the deviation of the sensor. In this paper, phenology information was added to the retrieval models and showed significant importance in the built models.

PI had been emphasized by many previous studies [29,60,61]. The merits of using PIs can be summarized as: (1) compared with other variables, they are easily acquired during the field experiments. Anyone can distinguish basic crop stages and record it by Zadok's scale stages. (2) PIs can also help to build models with less features, thus reducing the computation time while maintaining model accuracy (Figure 8). (3) PIs are expandable because of their consistency in one particular region [62]. For further expansion, PIs can be converted to other parameters, such as growing degree days (GDD), days of year (DOY), and days after sowing (DAS) [60]. All these parameters are either of a meteorological or time-series type, meaning that they can be determined even without agronomy expertise knowledge.

Zadok's stages or other stage codes, such as Feekes' scale or BBCH [63], are abbreviated designations. There are alternatives for PIs; furthermore, in addition to the GDD, DOY, or DAS mentioned above, leaf age is another index that can detail all growth situations. Otherwise, there could be a normalized index known as the relative growth stage (RGS) [61]. The potential utilization of these indicators still needs to be examined.

*4.3. Other Machine Learning Models Integrated with PIs*

This paper concluded that RF models combined with PI could yield accurate predictions of crop growth status parameters. The RF can only represent the decision-tree-type machine learning model [23,32]. The other types of machine learning models integrated with PIs need to be tested. For example, partial least squares regression (PLSR) and support vector regression (SVR) are the most commonly used machine learning algorithms in current remote-sensing data interpretation [19,33,56,64]. All combinations in this research were used to build models using different methods, and the different machine learning models had the same improved results when considering PIs in the models.

The results showed similar trends compared with RF models. C1 showed lower accuracy, while models that used C4 showed better performance, of which is the combination of C1 and PIs. Likewise, slight improvements were observed between C2 and C5 or C3 and C6. To summarize, the models consisting of band reflectance can be improved greatly, and the models consisting of VI can be improved slightly. The detailed results are presented in Tables 5 and 6:

**Table 5.** PLSR results for different combinations.

| | $R^2$ | | | | RMSE | | | | NRMSE | | | |
|---|---|---|---|---|---|---|---|---|---|---|---|---|
| | AGB | PNA | PNC | NNI | AGB | PNA | PNC | NNI | AGB | PNA | PNC | NNI |
| C1 | 0.62 | 0.67 | 0.42 | 0.68 | 2.25 | 33.96 | 0.33 | 0.16 | 14.65 | 14.13 | 16.44 | 15.15 |
| C2 | 0.81 | 0.78 | 0.76 | 0.74 | 1.57 | 27.9 | 0.21 | 0.14 | 10.24 | 11.61 | 10.38 | 13.61 |
| C3 | 0.81 | 0.78 | 0.78 | 0.75 | 1.58 | 27.53 | 0.20 | 0.14 | 10.31 | 11.45 | 10.00 | 13.28 |
| C4 | 0.74 | 0.69 | 0.67 | 0.67 | 1.86 | 33.02 | 0.25 | 0.16 | 12.15 | 15.93 | 12.36 | 15.34 |
| C5 | 0.82 | 0.78 | 0.74 | 0.74 | 1.56 | 27.75 | 0.21 | 0.14 | 10.15 | 11.55 | 10.83 | 13.57 |
| C6 | 0.82 | 0.79 | 0.75 | 0.76 | 1.56 | 27.06 | 0.21 | 0.14 | 10.16 | 11.26 | 10.62 | 13.18 |

**Table 6.** SVR results for different combinations.

| | $R^2$ | | | | RMSE | | | | NRMSE | | | |
|---|---|---|---|---|---|---|---|---|---|---|---|---|
| | AGB | PNA | PNC | NNI | AGB | PNA | PNC | NNI | AGB | PNA | PNC | NNI |
| C1 | 0.45 | 0.46 | 0.39 | 0.68 | 2.79 | 45.52 | 0.34 | 0.16 | 18.19 | 18.94 | 17.04 | 15.28 |
| C2 | 0.81 | 0.74 | 0.76 | 0.74 | 1.58 | 30.48 | 0.21 | 0.14 | 10.31 | 12.68 | 10.58 | 13.71 |
| C3 | 0.81 | 0.74 | 0.77 | 0.76 | 1.59 | 30.26 | 0.20 | 0.14 | 10.40 | 12.59 | 10.20 | 13.25 |
| C4 | 0.71 | 0.63 | 0.66 | 0.69 | 2.00 | 38.27 | 0.25 | 0.16 | 13.02 | 13.74 | 12.53 | 15.13 |
| C5 | 0.81 | 0.74 | 0.75 | 0.74 | 1.57 | 30.12 | 0.21 | 0.14 | 10.26 | 12.53 | 10.75 | 13.73 |
| C6 | 0.81 | 0.75 | 0.77 | 0.77 | 1.57 | 29.68 | 0.2 | 0.14 | 10.22 | 12.35 | 10.28 | 12.87 |

## 5. Conclusions

In this paper, we examined the linear relationships between remote-sensing indices and found that verified crop growth parameters were significantly affected by phenology. Statistically, phenology had a 16.04–49.87% contribution to different crop variables. Therefore, PIs were integrated into the random forest model to evaluate if it could improve the model's prediction ability. The results showed that most models can provide accurate predictions of the selected parameters with $R^2 > 0.7$, and PIs were important in the built models. The optimized RF model with best performance was analyzed using different nitrogen treatments or stages. The results were varied in different growth variables, with an average $R^2$ ranging from 0.61 to 0.75 in different stages. These results indicated that the phenology needed to be considered in future studies. Additionally, when using different types of remotely sensed data, PIs had different adaptations to the model effectiveness, and to better understand the insertion of PIs, more studies coupling with different PI datatypes need to be conducted. Future crop monitoring work needs to consider crop phenology and the possible ways of transforming phenology, as well as introducing other kinds of advanced machine learning regression methods into this subject.

**Author Contributions:** Conceptualization, S.H.; data curation, S.H. and Y.Z.; investigation, H.F.; methodology, Z.L. and J.C.; software, J.C.; validation, S.H. and Y.Z.; formal analysis, F.Z., H.Y. and X.M.; writing—original draft preparation, S.H. and Y.Z.; supervision and review, G.Y. and C.Z. All authors have read and agreed to the published version of the manuscript.

**Funding:** This study was supported by the Natural Science Foundation of China (42171303), Key scientific and technological projects of Heilongjiang province (2021ZXJ05A05), Chongqing Technology Innovation and Application Development Special Project (cstc2019jscx-gksbX0092, cstc2021jscx-gksbX0064), and the National Key Research and Development Program of China (2019YFE0125300).

**Data Availability Statement:** Not applicable.

**Acknowledgments:** The authors would like thank Hong Chang and Weiguo Li for acquiring data in the field experiments of this study.

**Conflicts of Interest:** The authors declare no conflict of interest.

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
