# Peer review of "Monitoring Key Wheat Growth Variables by Integrating Phenology and UAV Multispectral Imagery Data into Random Forest Model"

_remotesensing, doi:10.3390/rs14153723_

Round 1

Reviewer 1 Report

The paper is very good. I would like to see this paper public. 

Author Response

Response to Reviewer 1 Comments
Reviewer #1: The paper is very good. I would like to see this paper public.

Response: We appreciate so much for your fond, and we also endeavor to make the paper online soon.

Reviewer 2 Report

This research focused on estimating wheat properties (e.g., above ground biomass, plant nitrogen concentration) using UAV images, phenology indicators, and random forest. The topic is of interest, and research results can indicate the importance of phenology on estimating crop features. The manuscript is generally well-written. However, there are several major or minor concerns need to be addressed before considering publication. See details below.

·        L27, 3 types of data formed 6 combinations are unclear. Please specify.

·        L32, “availability” of PI? Or you mean importance?

·        L34,  “different stages or fertilizer 34 treatment”, this sounds will complicate the model.

·        44-45, language.

·        72-75, maybe good to break this long sentence to short ones.

·        100-101, sink to source of what?

·        196, please provide details of “standard reflectance panel”.

·        201-202, UAV data and PI are two types, or three types?

·        203-207, reference to Table 1 after this sentence.

·        Table 1, for PI, explain the “33, 47, 65…”

Author Response

Dear reviewer,

Thank you very much for your review. We fully considered your constructive comments and questions, and responded to your comments one by one in the following content.

For details please see the attachment.

Reviewer 3 Report

Information about the camera and its properties (resolution, pixel size) and GPS module should be illustrated.

Have you used ground control points (GCPs) for multi-date registration?

How do you extract VIs from images over sampling location? or is it a plot average?

Sec 3.1 I suggest to run Tukey's HSD test for statistical significance.

Sec 3.2 What is the rationale of selecting red band for assessing phenological proxy parameters?

Author Response

(The authors gave the same response as above.)

Round 2

Reviewer 2 Report

Authors revised the manuscript properly. No further comments.